# When did you leave home? Reconstructing juvenile life histories of coastal Washington adult coho salmon (*Oncorhynchus kisutch*) using otolith structure and chemistry

**Austin J. Anderson**⊙\*, **Andrew M. Claiborne, Marisa N. C. Litz, Lance Campbell**

Washington Department of Fish and Wildlife, Olympia, Washington, United States of America

\* austin.anderson@dfw.wa.gov

## Abstract

Coho salmon (*Oncorhynchus kisutch*) can express a range of juvenile life history strategies and understanding how they contribute to adult returns is critical for long-term harvest management and population resiliency in the light of environmental change. In this study we reconstructed the juvenile life histories of natural origin adult coho salmon returning to a lower and an upper basin tributary of the Chehalis River, in Washington state. First, we established the timing of otolith annulus formation for natural origin juveniles (n = 98) from the East Fork (EF) Satsop and the Newaukum River basins. We then used otolith microchemistry in conjunction with measurements of freshwater annulus location along the microchemistry transect from returning adults (n = 173) to describe the timing of freshwater emigration of fish from these two tributaries. Our results indicate that adult coho returning over two run years (2021 and 2022) exhibited seven unique successful juvenile life history strategies, three in the EF Satsop River and six in the Newaukum River. Emigration timing ranged from at least August through June and included individuals transitioning from age-0 to −1 and age-1 to −2. The majority (52−80%) of returning fish emigrated as age-1 smolts in the spring after their first otolith annulus had formed, however, 20−39% emigrated earlier, during active annulus formation (EF Satsop: October-April, Newaukum: October-December). A few individuals from the Newaukum River (2−3%) emigrated prior to the start of annulus formation, corresponding to a freshwater emigration before October. This is the first time successful juvenile migration strategies aside from spring age-1 smolts have been described for coho salmon in the Chehalis Basin. Identifying previously unrecognized life histories that make meaningful contributions to the spawning population highlights their impact on population stability and may help refine estimates of juvenile production, smolt-to-adult survival, and forecasting projections.

**Data availability statement:** All data for this study are available via a repository through the website Figshare at DOI: https://doi.org/10.6084/m9.figshare.c.7961852.

**Funding:** This work was funded by the Pacific Salmon Commission through the Southern Boundary Restoration and Enhancement Fund (Grant# SF-2022_SP-19) awarded to authors AA, AC, ML. The funders did not play any role in the study design, data collection/analysis, decision to publish, or preparation of the manuscript. https://www.psc.org/about-us/funding-opportunities/.

**Competing interests:** The authors have declared that no competing interests exist.

## Introduction

Diversity of life histories in animal populations is valuable because it spreads mortality risk across multiple behavioral strategies [1]. Diversity serves to buffer populations against harsh environmental conditions or other factors that have the potential to devastate a particular age class, phenotype, or behavioral group. In Pacific Salmon, life history variation has been shown to stabilize populations through time, where abundance or productivity of subpopulations may vary greatly from year to year, while variation in the overall population is buffered [2–4].

Coho salmon (*Oncorhynchus kisutch*) can express several different life history strategies, with the dominant age class generally varying by latitude [5]. Coho have been documented migrating to sea between age 0–4 and may spend anywhere from 6 to 30 months in the marine environment [5]. In Oregon and Washington, the dominant age class of returning adults is 1.1 (European Age Notation), which describes a fish that spent one winter in freshwater, migrated to sea in the spring of its second year, spent one winter in the ocean, and returned to freshwater to spawn in the fall of its third year [6,7].

While it has long been recognized that up to half of juvenile coho may perform their seaward migration at age-0 in the summer, fall, or winter of their first year [8–11], it was often assumed that these fry and parr migrants represent surplus production to a stream's carrying capacity and do not survive [12–14]. However, work in coastal Oregon and the Strait of Juan de Fuca has found that fall/winter age-0 migrants can survive to adulthood and comprised up to 50% of adult returns [11,15,16]. Many of these individuals use estuaries and adjacent streams as alternative winter rearing habitat before ultimately heading to sea the following spring [11,14].

Forecasting returns of wild coho in the Grays Harbor management unit (MU) in Coastal Washington state relies on accurate measurements of juvenile production (freshwater emigrants) and marine survival. Natural origin (hereafter wild) juvenile production and marine survival estimates have been generated annually since 1982 based on a Washington Department of Fish and Wildlife (WDFW) mark-recapture program of tagged juveniles that return as adults to Bingham Creek, a tributary to the East Fork Satsop River within the Chehalis River basin (Fig 1) [17–20]. The Bingham Creek program consists of a channel spanning juvenile fan trap and an adjacent adult weir trap. From approximately March through June, all coho moving downstream past the fan trap are captured, and smolts (individuals age ≥ 1) are marked with a coded wire tag (CWT) before being released. All returning adults are captured in the weir trap where hatchery coho are removed from the system and wild fish are passed above the trap. The proportion of untagged wild adults passing the Bingham trap often exceeds 50%, however, their origin is unclear. Current marine survival estimates are based entirely on CWT spring age-1 smolts, which are likely not accurate for any fish that migrate in the fall or winter outside of the trapping window [10,15,16,21]. If fall/winter migrants are surviving and contributing to adult returns, this has significant implications for estimating production and survival, which directly influences run size forecasting used for harvest management planning.

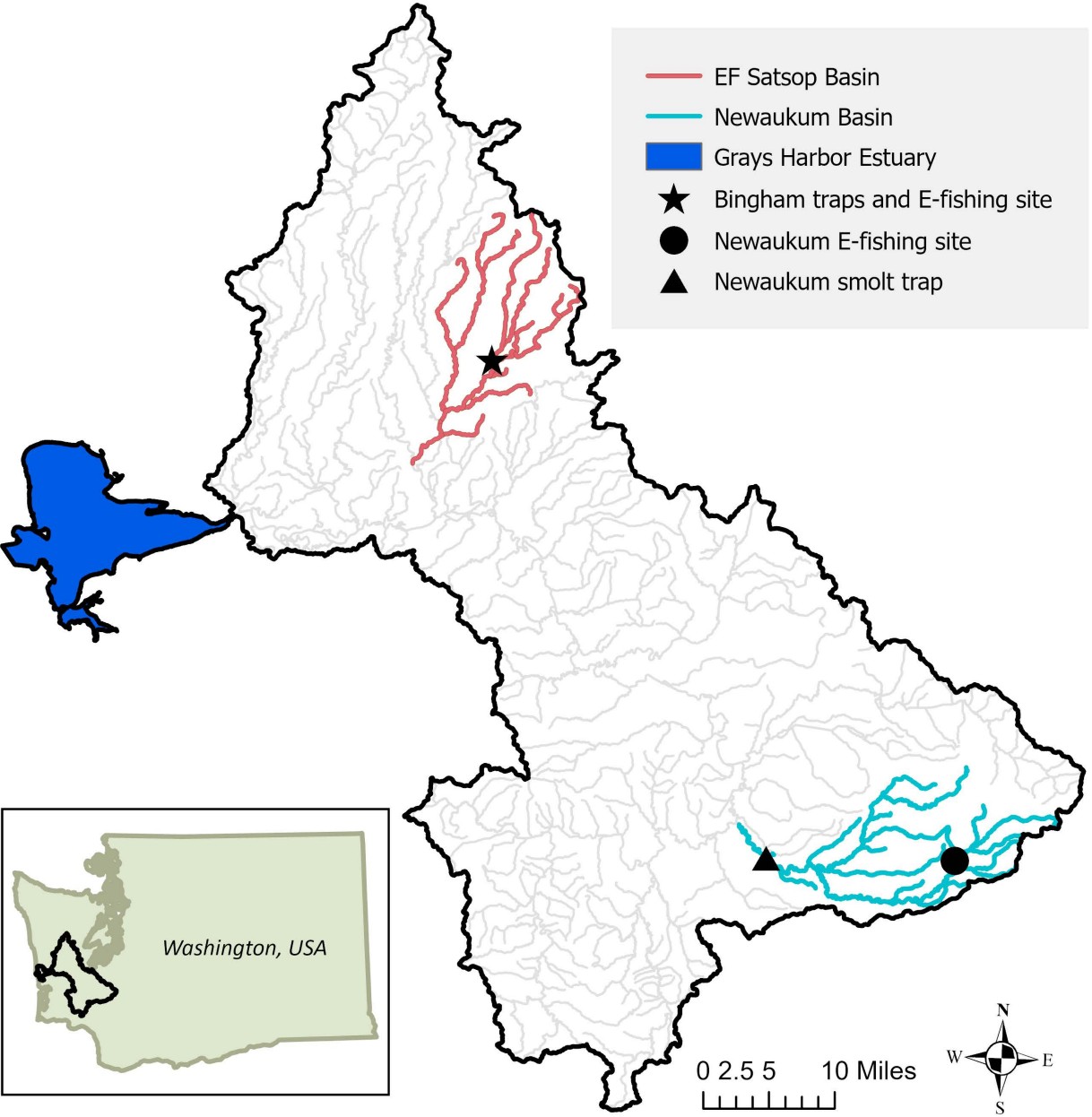

**Fig 1. Map of Chehalis River basin and Grays Harbor estuary.** Map services and data available from U.S. Geological Survey, National Geospatial Program [19,20]. (Map: L. Sikora, WDFW).

Sagittal otoliths (hereafter otoliths) are a pair of primarily calcium carbonate structures in the inner ear of fishes that are used for spatial orientation, balance, and hearing [22]. Otoliths grow throughout a fish's life by laying down concentric layers of new material and incorporate trace-elements from the surrounding water environment into their structure [23, 24]. Fish age can be determined by counting otolith annuli – yearly patterns that present as translucent rings in the otherwise opaque otolith structure and form during periods of slower growth. For temperate and polar zone fishes, annuli typically form in the winter when temperatures are cooler and feeding opportunities and metabolism are reduced [22,23]. In Pacific

salmon, otolith growth is directly related to fish growth (body length) [25], which allows for the back calculation of fish size at earlier points in their life [25–28].

Otolith microchemistry is a commonly used technique for reconstructing migration history in Pacific salmon. It involves performing mass spectrometry on a transect of otolith material from the core out to the edge and looking at the ratio of particular elements or isotopes along this transect [29–31]. In general, the ratio of strontium (Sr) to calcium (Ca) is positively correlated with water salinity, although some freshwater habitats have unusually high strontium concentrations [32–34]. Typically, low otolith Sr:Ca correspond to low Sr:Ca in freshwater habitats and high otolith Sr:Ca correspond to higher ratios found in estuary and ocean environments [29,32,33]. The combination of otolith chemistry, transect measurements of annuli locations, and fish size back-calculation models, may be used to estimate the size and timing of when fish emigrate from fresh to brackish/marine waters.

The primary objective of this study was to reconstruct the juvenile life histories of wild adult coho salmon returning to the Chehalis River basin, part of the Grays Harbor MU of Coastal Washington. To accomplish this objective, we first established the timing of otolith growth and winter annulus formation for wild juveniles from two tributaries to the Chehalis River. We then used otolith microchemistry in conjunction with annulus measurements from retuning adults to describe the timing of freshwater emigration of fish from these two tributaries over two return years. Identifying alternative life history strategies in wild coho that are not currently recognized provides insight into overall juvenile production and tests the assumptions of data currently used to inform fishery management in the Chehalis River basin.

## Methods

### Study populations

This study evaluated wild coho salmon returning to the Chehalis River basin, part of the Grays Harbor MU of Coastal Washington. We focused on two subbasins of the Chehalis River, the East Fork Satsop River (hereafter EF Satsop River), a lower basin tributary, and the Newaukum River, an upper basin tributary (Fig 1). These tributaries were selected for comparison because they both have robust wild coho populations, support spring juvenile smolt monitoring programs, and undergo extensive annual adult sampling. These tributaries are also part of a long-term stream temperature monitoring study from which we obtained monthly mean water temperatures from 2019–2022. Monthly temperature data was averaged across all years available to characterize thermal regimes in each basin [35].

### Sample collection

To obtain reference timestamps for annulus formation and baselines for otolith microchemistry we collected wild juvenile coho salmon every other month from August 2021 through June 2022 in both study basins (target sample size n = 10 per basin per sampling event). August, October, December, and February samples were collected via backpack electrofishing. April and June samples were collected via WDFW smolt trapping operations. All juvenile fish were humanely euthanized via overdose with tricaine methanesulfonate and fork length (FL) measurements, scale samples, and otoliths were taken from each individual. All juvenile fish were collected in accordance with WA state law RCW 77.12.071 which permits WDFW employees to collect samples of tissue, or other bodily parts of fish from state waters. All juvenile sampling and euthanasia practices adhered to the ethical standards laid out in the American Fisheries Society Guidelines for the use of Fishes in Research [36].

Returning wild adult coho from each basin were sampled from two return years, 2021 and 2022. In the Newaukum River, unmarked coho were sampled when encountered during basin-wide fall spawning ground carcass surveys incorporating efforts on foot and floated by raft or pontoon boat. In the EF Satsop River, samples were collected from wild coho spawned for the integrated broodstock program at Bingham Creek Hatchery (females only), located on the EF Satsop River approximately 1000 ft upstream of its confluence with Bingham Creek. Additional EF Satsop samples were collected opportunistically from recreational fishing effort or during floated spawning ground carcass surveys downstream of the

 

hatchery. Three Satsop fish were sampled outside of the East Fork drainage; one on the West Fork and two on the main-stem Satsop River. FL measurements, scale samples, and otoliths were taken from each individual with a goal of at least n = 50 per basin per return year. To avoid biasing our EF Satsop samples towards spring outmigrants from the Bingham Creek juvenile fan trap CWT program, only unmarked, non-CWT fish were included in this study. Any jacks or fish classified as hatchery origin via scale analysis were excluded. Scales from wild Grays Harbor coho have a smaller and more defined first annulus than those of hatchery coho [37]. Hatchery coho scales also consistently display freshwater growth checks (S1 Fig). All adult coho were either naturally deceased or were humanely euthanized for WDFW hatchery broodstock practices via percussive stunning prior to sampling by the authors. RCW 77.12.071 permits WDFW employees to collect samples of tissue, or other bodily parts of fish from state waters. Grays Harbor coho salmon are not listed under the U.S. Endangered Species Act, so no federal permit was required. All practices adhered to the ethical standards laid out in the American Fisheries Society Guidelines for the use of Fishes in Research [36].

## Otolith preparation, chemical analysis, and life history reconstruction

In preparation for chemical analysis, juvenile and adult coho sagittal otoliths were cast in Crystal Bond 509 TM, thin sectioned in the sagittal plane, and polished as described in [38]. We measured otolith Sr and Calcium (Ca) using a Thermo X series II inductively coupled plasma mass spectrometer coupled with a Photon Machines G2 193 nm excimer laser (i.e., LA-ICPMS) at the Keck Collaboratory for Plasma Mass Spectrometry at Oregon State University. Laser ablation scans were completed from the primordia of each otolith at approximately 25º in the dorsal posterior quadrant to the edge (Fig 2A). The laser was set at a pulse rate of 8 Hz traveling across the sample at 5 μm/s with a spot size of 30 μm. Normalized ion ratios of Sr:Ca were converted to elemental concentration using a glass standard from the National Institute of Standards and Technology (NIST 610) and finally converted to molar ratios for analysis similar to [39]. Once converted, we identified the inflection point where the Sr:Ca ratio spiked and surpassed 1.5 mmol/mol [33,34], an indication that a fish has transitioned from freshwater to the marine environment (hereafter freshwater emigration).

## Age estimation and timing of freshwater annulus development

Following microchemical analysis we imaged all juvenile and adult otoliths under reflected light. Two readers independently estimated fish ages using the images to visually inspect otolith structure and count annuli. Fish were assumed to have January 1st birthdays for ageing purposes. This was an important consideration when assigning age to juveniles whose capture dates (Aug-June) spanned calendar years and the winter period of annulus formation. Based on the number of otolith annuli, annulus proximity to otolith edge, and the date of capture, juveniles were assigned a single, integer age (0, 1, or 2 years-old). For adults, we counted the total number of annuli present to assign a total age (e.g., 2 or 3 years-old). Readers resolved any discrepancies in age by reviewing images together and coming to a consensus. Image Pro Plus 7.0 was used to measure each otolith from the core to the start and end of each annulus, and to the outer otolith edge following the laser ablation transect (Fig 2A).

Juvenile otolith images were also used to assign one of three annulus development stages to each individual. The three possible development stages were "before", "during" or "after" annulus formation (Fig 2A). The "before" annulus development stage was characterized by opaque material extending to the otolith edge. The "during" stage was characterized by having opaque structure surrounded by transparent annulus material extending to the otolith edge. The "after" stage was characterized by having a fully formed transparent annulus surrounded by opaque material extending to the otolith edge. Readers then resolved any discrepancies in annulus development types by reviewing images together and coming to a consensus.

## Freshwater emigration timing and life history type classification

Adult coho life history types were classified using fish age and the timing of freshwater emigration. The relative timing of freshwater emigration was identified by comparing the Sr:Ca freshwater emigration location on the laser transect to the

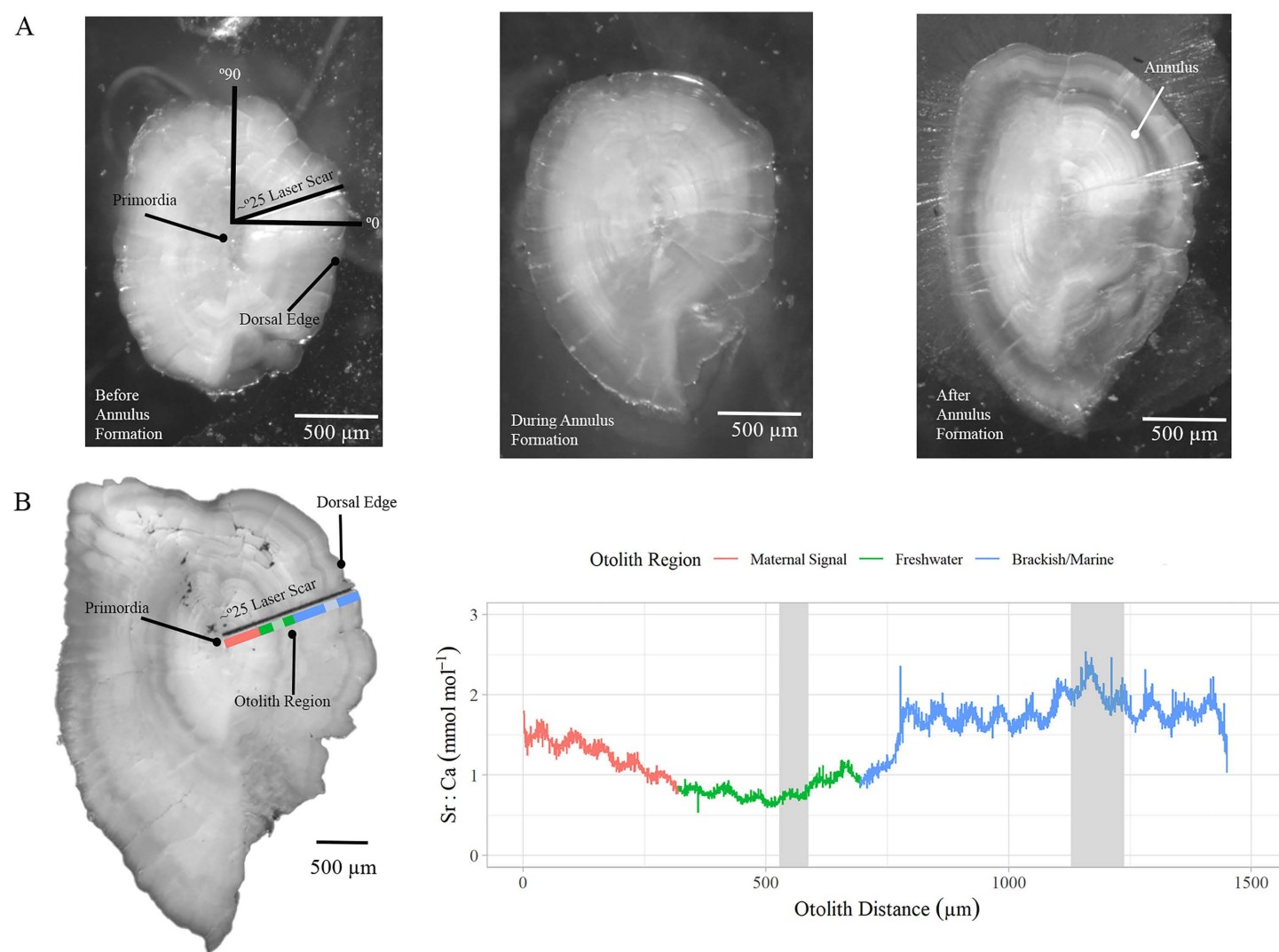

**Fig 2. Otolith annulus development stages and laser ablation transect example.** (A) Juvenile coho salmon (*Oncorhynchus kisutch*) otoliths from before, during, and after (left to right) annulus formation. (B) Adult coho salmon otolith and the corresponding Sr:Ca profile from the primordia to dorsal edge for an individual classified as life history type I. The red line indicates the maternal region, green indicates freshwater, blue indicates brackish marine Sr:Ca values while grey boxes are the otolith annuli.

otolith's juvenile annulus measurements to determine if emigration occurred before, during, or after annulus formation (Fig 2B). A life history was defined as a unique combination of juvenile emigration age, emigration timing, and the total number otolith annuli present (e.g., a fish that emigrated at age-1 after its first annulus formed with a total age of 2).

## Fork length estimation

The majority of juvenile coho sampled were measured for FL immediately upon capture and then frozen to be processed after all sampling events had been completed. For two of the sampling events juveniles were not measured before being frozen. To account for this, we measured all juvenile fish at the time of processing after they had been thawed and created a liner model to estimate the fresh FL in mm of fish that were not measured at the time of capture: Fresh FL = 1.0613*Frozen FL − 2.53, $R^2$ = 0.997.

We estimated the FL of returning adult salmon at different points in their life (locations on otolith) using the relationship between FL and otolith radius [27,28]. To generate these estimates, we first determined if separate models were needed for the two populations by testing if the slopes of juvenile FLs versus otolith radii differed significantly between populations, they did not (ANOVA, p = 0.16). Using our juvenile coho FL measurements and otolith samples, we then created a linear regression model of juvenile FL at capture versus otolith radius:

$$FL_j = 0.170838 * OR_j - 20.335212, R^2 = 0.753$$

Where $FL_j$ is the FL in mm at the time point of interest, and $OR_j$ is the otolith radius in µm at the time point of interest. We applied this model to our adult coho samples to back-calculate FL at three locations on their otoliths: the point of freshwater emigration, the start of any juvenile otolith annuli, and the end of any juvenile otolith annuli. These were time points of interest because we used the location of freshwater emigration in relation to the juvenile annulus to classify life history types.

### Data analysis

To compare the proportions of juvenile life histories in returning adults we used the Fisher's Exact Test. Specifically, we compared the most numerically dominant life history type to the sum of alternative types between years within basins and between basins for each year. We used the Kruskal-Wallis Test to test for overall differences in back-calculated FL at freshwater emigration between basin and year combinations. We also used the Kruskal-Wallis Test to test for differences in back-calculated FL at freshwater emigration, the start of first annulus development, and at the end of first annulus development between dominant life history types. Pairwise comparisons of FL at freshwater emigration were made between basin and year combinations using the Pairwise Wilcoxon Test with p-values adjusted for multiple comparisons using the Holm method. Similarly, we used the Pairwise Wilcoxon Test to make within basin pairwise comparisons between dominant life history types and year combinations of FL at freshwater emigration, at the start of first annulus development, and at the end of first annulus development. Analyses were completed in R [40] and utilized package "rstatix" [41]. Figures utilized packages "ggplot2" [42], "gridExtra" [43], and "plyr" [44].

## Results

### Timing of juvenile annulus development

We sampled a total of 98 juvenile coho (EF Satsop n = 50, Newaukum n = 48). Wild juvenile coho salmon sampled in August, October and December were age-0, except for two age-1 individuals (S2 Fig.) captured in the Newaukum River in December (Table 1). Juvenile coho salmon sampled in February, April, and June of 2022 were age-1, with the exception of fish in the Newaukum smolt trap in June, where no age-1 smolts were encountered (Table 1). We observed that none of the juvenile coho salmon sampled had previously emigrated from freshwater based on otolith Sr:Ca. In the EF Satsop and Newaukum Rivers, mean FL of sampled fish generally increased from August through June, although little change occurred between the October-December sampling events. Rapid increases in FL were observed from December through April in both study basins (Fig 3A).

Otolith annulus formation began at approximately the same time in both basins – between the August and October sampling events (Fig 3B). However, the duration of annulus formation was longer for fish captured in the EF Satsop than Newaukum River (Fig 3B). In the EF Satsop Basin, we observed juvenile otoliths during the process of annulus formation from August into the following April, although some fish (20%) had completed annulus formation by February, and most (80%) had completed formation by April. In the Newaukum Basin, we only observed fish during annulus formation in October and December; by the February sampling event 100% of sampled Newaukum fish were captured after annulus formation was complete (Fig 3B). In general, freshwater annulus translucent zones of Newaukum River otoliths had more contrast with opaque zones than those of EF Satsop otoliths (S3a in S3Fig). Readers noted this same pattern when viewing and imaging adult otoliths (S3b in S3 Fig).

**Table 1. Juvenile coho salmon ages and sample size by month and basin.**

| Month | EF Satsop | | | Newaukum | | |
|---|---|---|---|---|---|---|
| | Age-0 | Age-1 | Total | Age-0 | Age-1 | Total |
| Aug | 11 | | 11 | 10 | | 10 |
| Oct | 10 | | 10 | 10 | | 10 |
| Dec | 7 | | 7 | 9 | **2** | 11 |
| Feb | | 5 | 5 | | 7 | 7 |
| Apr | | 10 | 10 | | 10 | 10 |
| Jun | | 7 | 7 | | | 0 |

We assume a January 1st birthdate for ageing purposes (i.e., a fish that was age-0 in December 2021 would be age-1 in January 2022). Bold number indicates the age-1 juveniles captured in the Newaukum River in December forming their second annulus (S2 Fig).

Mean temperature data showed water temperature regimes varied between the two basins (Fig 3A). We observed greater maximum summer temperatures and lower minimum winter temperatures in the Newaukum compared to the EF Satsop Basin. In both systems, the onset of annulus formation was concurrent with the decline of summer temperatures (Fig 3). The increase in temperature between February and April in both systems was more closely aligned with completion of winter annuli for juveniles in the EF Satsop than in the Newaukum River.

### Life history types and juvenile size of returning adults

We processed a total of 173 wild adult coho otoliths (Satsop n = 69, Newaukum n = 104). Across basins and return years we observed seven unique successful juvenile life history types in returning adults based on age and freshwater emigration timing (Table 2, Fig 4). Most (97%) individuals had two total annuli and emigrated from freshwater before, during, or after formation of the first annulus as they transitioned from age-0 to age-1. Depending on run year and basin, 52−80% of individuals were classified as type I, where fish emigrated from freshwater after their first annulus had developed (spring age-1 smolts). 20−39% were classified as type II, where fish emigrated during active formation of their first annulus (winter age-0/1 migrants). In the Newaukum River, 2−3% of fish were classified as type III and emigrated from freshwater prior to the start of their first annulus forming (fall age-0 migrants). No fish in the EF Satsop River were classified as type III. Additionally, we observed four life histories (type IV, V, VI, and VII) characterized by the presence of three total annuli (3%, 5/173). Life history type IV was characterized by emigration from freshwater during formation of the second of three annuli (winter age-1/2 migrants); two fish from the Newaukum followed this pattern. Life history type V was characterized by freshwater emigration after formation of the second annulus (spring age-2 smolt); one fish from the EF Satsop River followed this pattern. Life history types VI and VII emigrated from freshwater during and after their first annulus respectively but appeared to form a second annulus in the brackish/estuary environment before forming a third annulus at sea (S4 Fig.). We assumed the second annulus formed in the estuary because chemistry indicated it occurred in marine waters, but its characteristics were similar to those of a freshwater annulus – located close to the core in the region readers typically observe juvenile coho freshwater annuli and distinctly different from the larger subsequent ocean annulus. Fisher Exact tests did not identify a significant association between basin and the proportion of life history type-I to all other types (II-VII) in 2021 (p = 0.31, odds ratio = 1.84) or 2022 (p = 0.08, odds ratio = 2.41), nor did they identify a significant association between year and life history proportions within basin (EF Satsop: p = 0.57, odds ratio = 1.49; Newaukum: p = 0.11, odds ratio = 1.96).

On average, size at freshwater emigration was 111 ± 9 mm FL (mean ± sd) in the EF Satsop River and 101 ± 14 mm FL in the Newaukum River across all life history types. Size at freshwater emigration significantly varied between basin and year combinations (Kruskal-Wallis tests, p < 0.001, H = 24.2, df = 3). Pairwise Wilcoxon comparisons indicated that in return year 2021 and 2022, size at freshwater emigration was significantly smaller for fish returning to the Newaukum

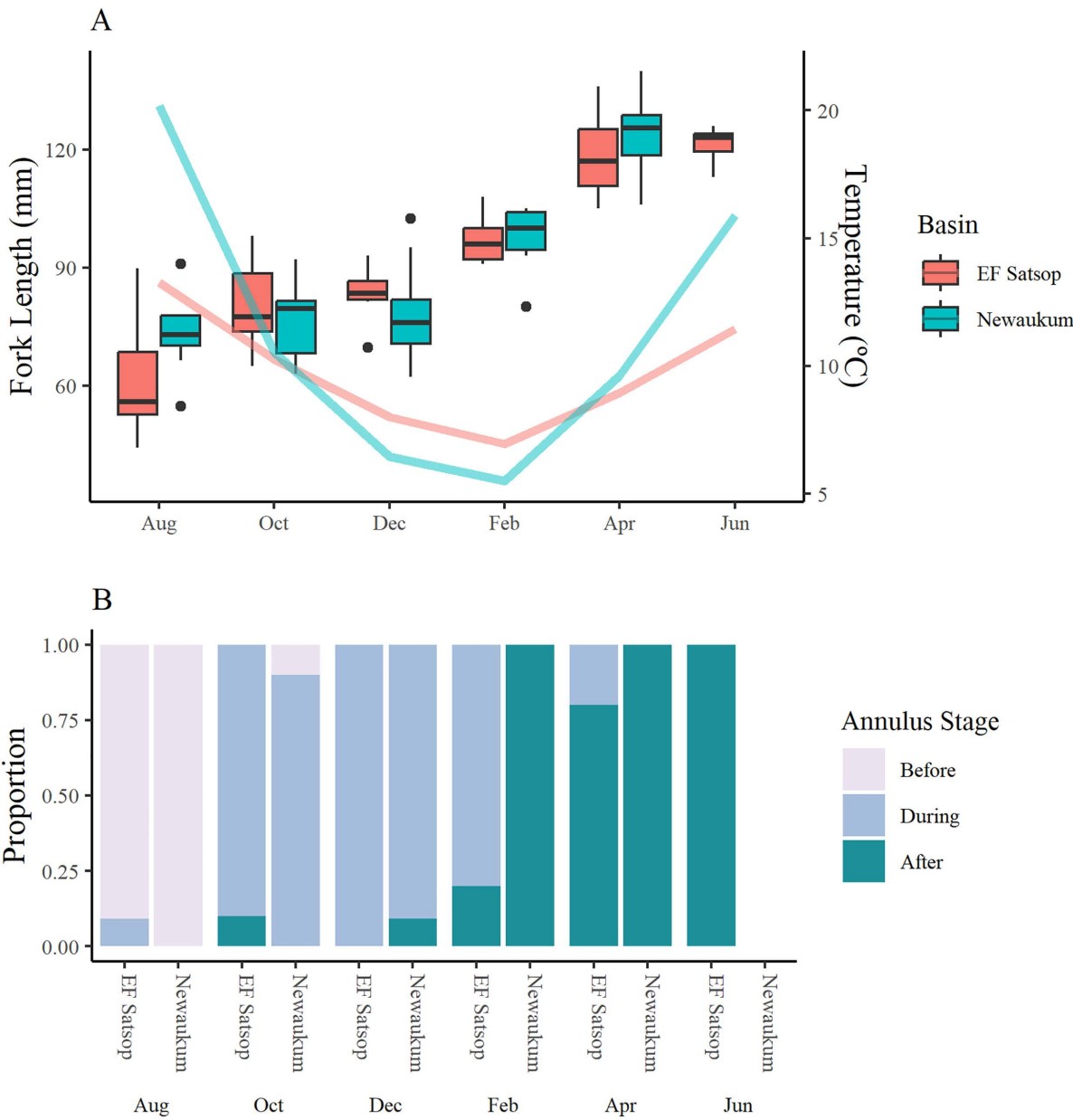

**Fig 3. Juvenile coho size, annulus development, and water temperature through time.** (A) Mean monthly stream temperature (lines, 2019-2022 average) and juvenile coho salmon (*Oncorhynchus kisutch*) bi-monthly median fork length (box and whisker) from August 2021 to June 2022 in the EF Satsop and Newaukum Basins; (B) corresponding proportion of annulus development stage by month and tributary.

compared to the EF Satsop River (Tables 3,4). However, there was no difference in size at freshwater emigration between run years for either basin (Tables 3,4). Overall, size at freshwater emigration significantly varied between life history type-I and type-II (Fig 5, Kruskal-Wallis test, p < 0.001, H = 28.8, df = 1). Pairwise comparisons for each basin indicated that in the Newaukum River, size at freshwater emigration was significantly smaller for fish classified as life history type-II versus type-I in 2021 and 2022. There was not a significant difference in freshwater emigration size between life history types in the EF Satsop River for either year (Fig 5, Tables 3–4).

**Table 2. Adult coho salmon sample size and life history type proportions by basin and return year.**

| Basin | n | Return Year | Juv. Age 0/1 | | | Juv. Age 1/2 | | | |
|---|---|---|---|---|---|---|---|---|---|
| | | | LH I | LH II | LH III | LH IV | LH V | LH VI | LH VII |
| EF Satsop | 25 | 2021 | 80.0% | 20.0% | – | – | – | – | – |
| | 44 | 2022 | 72.7% | 25.0% | – | – | 2.3% | – | – |
| | 69 | **Overall** | 75.4% | 23.2% | – | – | 1.4% | – | – |
| Newaukum | 60 | 2021 | 68.3% | 26.7% | 3.3% | – | – | 1.7% | – |
| | 44 | 2022 | 52.3% | 38.6% | 2.3% | 4.5% | – | – | 2.3% |
| | 104 | **Overall** | 61.5% | 31.7% | 2.9% | 1.9% | – | 1.0% | 1.0% |

Life history (LH) types I, II, and III are defined as adult coho with two total annuli who emigrated from freshwater after, during, or before formation of their first annulus, respectively (juvenile age 0/1). Type IV and V are defined as adult coho with three total annuli that emigrated from freshwater during, and after formation of their second annulus, respectively (juvenile age 1/2). Type VI and VII are defined as adult coho with three total annuli that emigrated from freshwater during, and after formation of their first annulus, respectively, but formed a second annulus while presumably in the brackish/marine environment before migrating to the ocean (juvenile age 1/2).

Overall, size at the beginning and end of the first annulus significantly varied between life history type-I and type-II (Fig 5, Kruskal-Wallis tests: start of annulus $p < 0.001$, $H = 14.9$, $df = 1$; end of annulus $p < 0.001$, $H = 24.5$, $df = 1$). Pairwise comparisons indicated that for the EF Satsop River size at the beginning and end of the first annulus was significantly larger for fish classified as life history type-II versus type-I in 2021 and 2022 (Fig 5, Tables 3–4). There were no significant differences in size at the beginning of the first annulus between life history types in the Newaukum River, however type-II fish were larger than type1 fish at the end of the first annulus for the run year 2022 group (Fig 5, Tables 3–4). The majority of type II fish (81% EF Satsop, 85% Newaukum) emigrated during the second half of annulus formation (Fig 5, S5 Fig.).

## Discussion

In this study we used a combination of juvenile and adult collections, otolith chemistry, and annulus measurements to describe the juvenile life histories of adult coho salmon returning to two tributaries of the Chehalis River over two run years. We found freshwater annulus formation began at a similar time in both tributaries (by October) but finished earlier in the Newaukum (by February) than the EF Satsop River (by April). This is consistent with the extended duration of cool stream temperatures in the groundwater-fed EF Satsop River compared to the rain-driven Newaukum River that warms rapidly in the spring. Our results demonstrate that Chehalis Basin coho salmon exhibit at least seven unique successful freshwater emigration strategies. We found evidence that surviving adults entered brackish/marine waters of the estuary from August through the following June as they transitioned from age-0 to −1 and from age-1 to −2, highlighting the importance of intact estuary rearing habitat for early migrants to utilize before ultimately heading to the ocean. This is the first time juvenile life histories other than spring age-1 smolts have been described for adult coho in the Chehalis Basin. Identifying successful life history strategies that are not currently recognized will help refine estimates used for fishery management (juvenile production and smolt-to-adult survival) and improve our understanding of the relationship between life history diversity and population stability across the Grays Harbor Basin.

Approximately one third (33%, 57/173) of returning adults across the two subbasins and return years exhibited non-spring age-1 smolt life histories (types II-VII). Past work has shown the presence of age-0 coho salmon in the Grays Harbor estuary each month between March and September, often encountered in higher densities than their age-1 counterparts in beach seine sampling [45]. Previous studies have demonstrated contributions of alternative juvenile life histories to adult returns in other Oregon and Washington watersheds. For example, Bennett et al. [15] passive integrated transponder (PIT) tagged nearly 26,000 juvenile coho over 7 years in the Twin Rivers on the Olympic Peninsula, Washington. They detected 86 tagged fish returning as adults, 32 (37%) of which had emigrated from freshwater in the

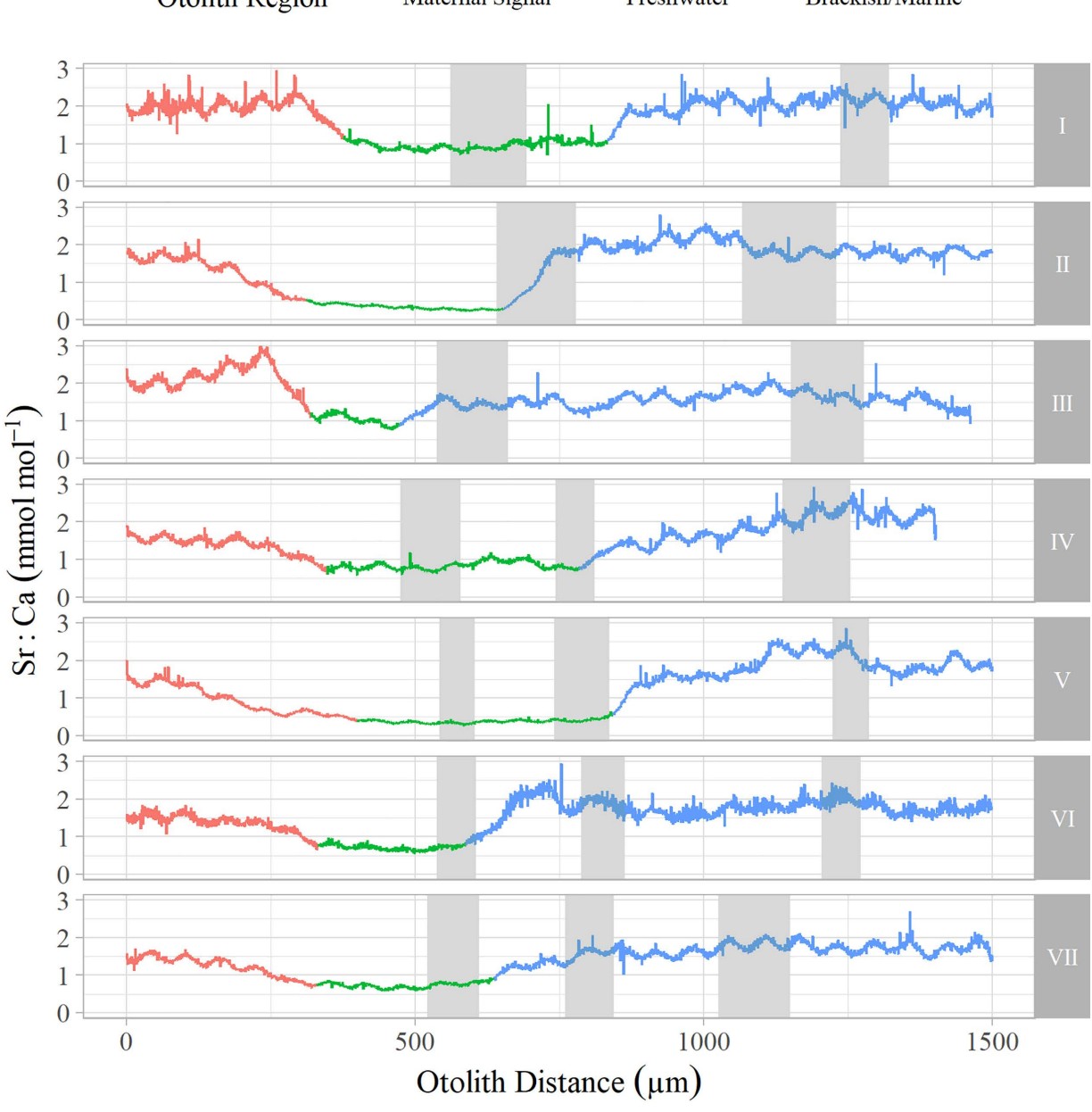

**Fig 4. Adult coho life history types.** Otolith Sr:Ca profiles from the primordia to dorsal edge showing each life history type from returning adult coho salmon (*Oncorhynchus kisutch*) in the EF Satsop and Newaukum Rivers. Otolith regions are shown where the red the line indicates the maternal region, green indicates freshwater, blue indicates brackish marine based on Sr:Ca values. Grey boxes are the otolith annuli.

fall or winter (before Feb 28). Jones et al. [11,16] used a combination of otolith chemistry and PIT tagging in the Salmon River, Oregon and observed 21−58% of returning adults to have emigrated from freshwater as age-0 fry or parr across 7 brood years. In addition to demonstrating diverse juvenile life histories in adult returns, both projects showed complex rearing strategies by early migrants. In the Salmon River, Jones et al. [11] found significant winter estuary use by early migrants (estuary residence: 17−311 days), and both Jones et al. [11,16] and Roni et al. [10] (precursor to Bennett et al.

**Table 3. Summary of pairwise Wilcoxon tests.**

|  | Comparison | P-value | Test statistic |
|---|---|---|---|
| Size at Freshwater Emigration | EF Satsop 2021 vs Newaukum 2021 | **0.002** | 1124 |
|  | EF Satsop 2022 vs Newaukum 2022 | **0.008** | 1341 |
|  | Newaukum 2021 vs Newaukum 2022 | 0.97 | 1214 |
|  | EF Satsop 2021 vs EF Satsop 2022 | 0.97 | 540 |
|  | EF Satsop 2021 LH I vs LH II | 0.77 | 71.5 |
|  | EF Satsop 2022 LH I vs LH II | 0.77 | 220 |
|  | Newaukum 2021 LH I vs LH II | **0.017** | 490 |
|  | Newaukum 2022 LH I vs LH II | **<0.001** | 338 |
| Size at Start of Annulus | EF Satsop 2021 LH I vs LH II | **0.041** | 13 |
|  | EF Satsop 2022 LH I vs LH II | **0.032** | 78 |
|  | Newaukum 2021 LH I vs LH II | 0.468 | 239 |
|  | Newaukum 2022 LH I vs LH II | 0.182 | 119 |
| Size at End of Annulus | EF Satsop 2021 LH I vs LH II | **0.001** | 3 |
|  | EF Satsop 2022 LH I vs LH II | **<0.001** | 43 |
|  | Newaukum 2021 LH I vs LH II | 0.144 | 210 |
|  | Newaukum 2022 LH I vs LH II | **0.018** | 91 |

Summary of pairwise Wilcoxon tests comparing median back calculated fork lengths at freshwater emigration, start of first annulus formation, and end of first annulus formation from adult coho otoliths between tributaries and run years and life history types I and II. Bold indicates significance at the alpha = 0.05 level. P-values were adjusted using the Holm method.

**Table 4. Back calculated sizes of life history type I and II fish.**

|  | | Fork Length (mm) | | |
|---|---|---|---|---|
|  | LH Type | Start 1st Annulus | End 1st Annulus | FW Emigration |
| EF Satsop | I | 73 (11) | 95 (12) | 112 (9) |
|  | II | 86 (13) | 114 (9) | 106 (9) |
| Newaukum | I | 72 (11) | 90 (12) | 106 (14) |
|  | II | 78 (8) | 98 (9) | 92 (8) |

Mean (Standard deviation) back calculated fork lengths (mm) of life history (LH) type I and II coho at the start of first annulus formation, end of first annulus formation, and at freshwater (FW) emigration estimated from adult coho otoliths. Fork lengths were averages for fish collected in both run year 2021 and 2022.

[15]) observed fall age-0 freshwater emigrants re-entering freshwater to overwinter, sometimes in different streams, before initiating ocean migration in the spring.

Although spring age-1 migrants accounted for the majority of returning adults, we observed variability in the proportion of alternative life histories, which represented a non-trivial share of fish in both run years and basins (20−48%, LH types II-VII). Currently, marine survival estimates for coho salmon in the Grays Harbor MU are based on adult CWT recoveries (fisheries and spawning escapement) from a known number of spring age-1 smolts tagged between March and June at the channel spanning fan trap on Bingham Creek in the EF Satsop basin [18]. These survival estimates are extrapolated to other Washington coast MUs for run size forecasts. The Bingham Creek life cycle monitoring program aims to tag 100% of age-1 coho passing the trap, although fish migrating outside of the trapping window and fry migrating during active trapping both avoid tagging. While the CWT-based marine survival estimate represents the spring age-1 smolts, marine survival of fall/winter migrants is likely lower [10,11,15]. Given the observed prevalence of fall/winter migrants in the

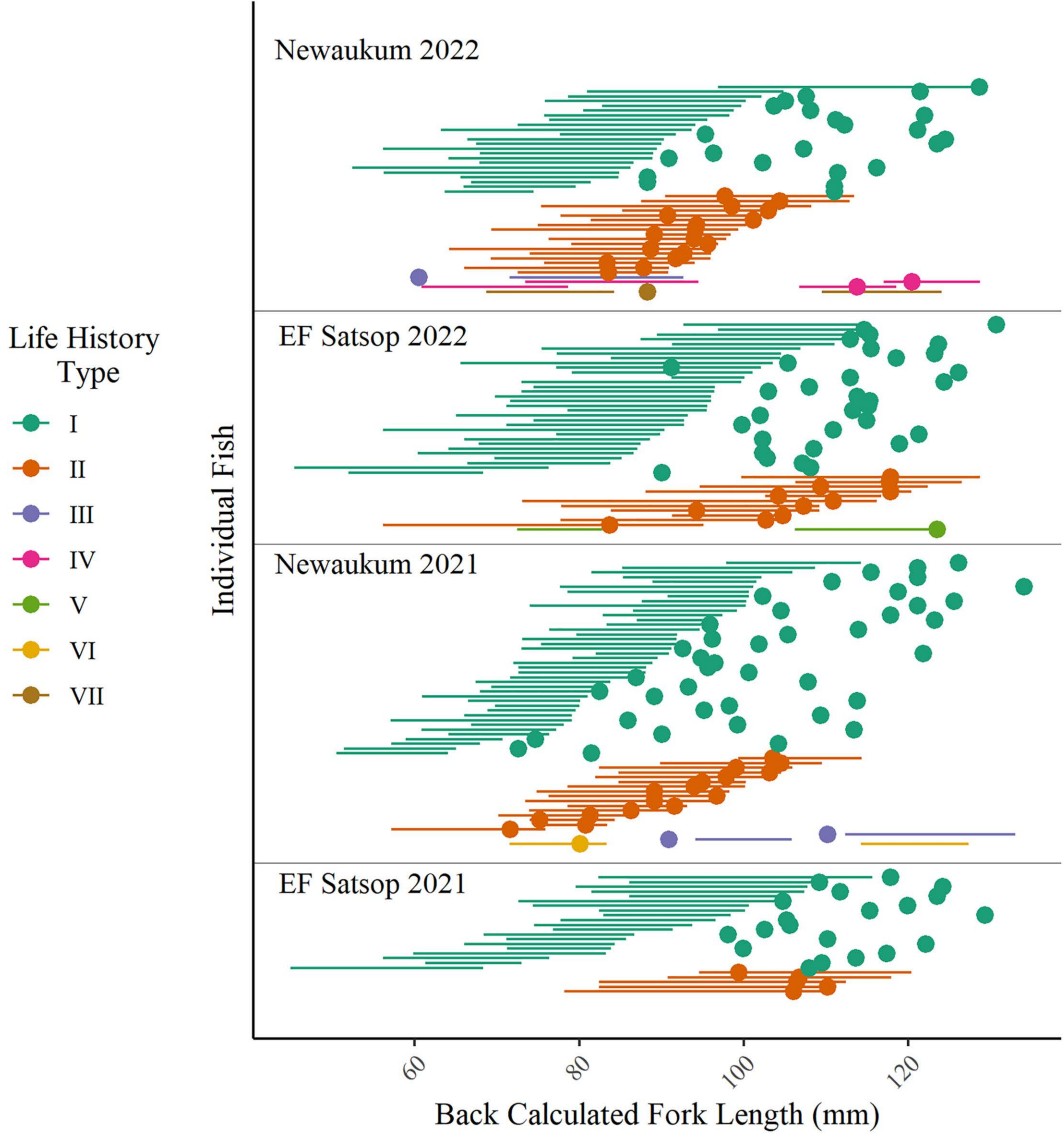

**Fig 5. Size at freshwater emigration in relation to juvenile annulus development.** Individual back-calculated fork length at freshwater emigration (dots) based on Sr:Ca and at each juvenile annulus (lines) for returning adult coho salmon (*Oncorhynchus kisutch*) in the EF Satsop and Newaukum Rivers. Individuals are ordered by life history type and by size at the end of their freshwater annulus.

returning adult spawners, these life histories may be a critical component to improving marine survival inputs to the run size forecasts.

Previous studies have noted that failing to account for early migrants will result in an underestimate of juvenile production as measured during the spring outmigration period [10,21]. To avoid this mistake, fishery scientists have historically estimated the total number of juveniles (CWT and untagged) emigrating from the Grays Harbor MU using the ratio of CWT:total wild adults captured in the Grays Harbor fishery. Total adult run size is generated by multiplying this back-calculated juvenile estimate by the Bingham CWT marine survival. However, this method has consistently predicted adult run sizes >50% greater than the actual return (Litz unpublished data), leading fishery scientists to investigate additional forecasting methods. Our study reveals a critical adjustment is needed. Specifically, while the back calculated smolt

abundance estimate accounts for both age-0 and age-1 migrants, the CWT-based marine survival estimate represents only spring age-1 smolts and likely overestimates survival of any fall/winter age-0/1 migrants [10,11,15]. Fall/winter migrants have been documented making up >50% of outmigrants in Oregon and Washington populations but are not typically observed in such high proportions in the adult returns [8,10,11]. We posit that the Grays Harbor run size forecast discrepancy can be explained by early migrants (i.e., type II), with lower marine survival than spring age-1 smolts, accounting for a significant proportion of the juvenile coho production in the system.

An additional unknown surrounds untagged fish returning to Bingham Creek. For decades, the number of untagged wild adult coho returning to the Bingham weir has approached or exceeded the number of CWT wild fish despite the program tagging all emigrating age-1 smolts (e.g., 51% untagged wild adults returning in RY 2021, 54% in RY 2022, Litz unpublished data). We reason there are three primary pathways that would give rise to these untagged fish: A) straying adults originally from other parts of the Satsop/Chehalis Basin. B) Fish that pass the active trap as fry/parr and do not receive a CWT. These fish may either rear downstream in freshwater (type I, IV, V, or VII) or proceed to the estuary (type III). Or C) winter migrants (type II or VI) that leave the basin before the trap is installed in the spring. While it is likely that all three pathways contribute to the untagged wild fish observed at the weir, this study provides insight suggesting type II winter migrants make up a substantial portion of those returns.

Because of a potential source of bias in our sampling design, the true proportion of type I migrants in the EF Satsop River may have been larger than we reported. We sampled only wild, non-CWT adults, primarily near the confluence of Bingham Creek and the EF Satsop, which could have biased against sampling wild CWT spring outmigrants and inflated our proportion of winter migrants. However, the majority (75%) of Satsop coho were found to be type I spring smolts, and no adults from the EF Satsop Basin were collected above the Bingham Creek trap site. This suggests our EF Satsop type I fish originated from outside the Bingham drainage or followed pathway B above, passing the trap as fry/parr and overwintering downstream in freshwater.

We identified four uncommon life history types characterized by three total annuli. Two of these life histories appeared to have formed (or partially formed) one annulus in freshwater, a second in the estuary, and a third in the ocean (types VI and VII). We assumed that the second annulus for these fish formed in the estuary because chemistry data indicated they occurred in the marine environment, but their location and spacing presented like freshwater annuli (S4 Fig.). It is possible what was identified as the second annulus was actually the second half of a split annulus [22]. In this scenario the first half of the annulus could have formed in freshwater, interrupted by rapid growth upon entry into the estuary, only to resume forming later that same winter. Alternatively, the second annulus for these fish could have been a strong growth check that occurred during the transition from estuary residence to true seaward migration [22,46]. Unfortunately, with the data we have we cannot know for certain.

Three annulus adult coho have been consistently present at low rates in the Grays Harbor CWT data (<1% of adult returns, M. Zimmerman pers. comm.). Additionally, estuarine overwintering during the second juvenile winter after spending the first in freshwater has been previously documented in Alaskan coho populations on the Kenai Peninsula [47]. However, freshwater age 2 coho are common in Alaska [5], so these fish are more comparable to the Type-III fall age-0 migrants in our study. Alternatively, other large estuary systems like the Puget Sound contain "resident" coho, subsets of populations that spend the entire marine phase of their life history within the bounds of the estuary [48]. The observed variation in juvenile size, rearing behavior, and migration timing demonstrates a greater diversity of successful coho life histories in the Grays Harbor watershed than was previously understood.

We documented differences in size at freshwater emigration between basins and life history types. Among *Oncorhynchus spp.*, differences in migratory size and timing can be the result of variability in freshwater conditions between tributaries or stream reaches, despite common downstream conditions [10,49]. Coho salmon have protracted spawn seasons with variable timing within and among tributaries of the Chehalis Basin (Nov-Feb, [50]), which would suggest diversity in emergence timing and spatiotemporal variation in growth [51]. In the present study, Newaukum River juveniles emigrated from freshwater at

smaller sizes than fish returning to the EF Satsop River. We hypothesize this to be a result of the earlier emigration window for type II fish in the Newaukum, combined with temperature regime differences between the basins. The EF Satsop Basin is largely groundwater driven and had less variation between summer and winter temperatures over the study period than the Newaukum, a more rain driven system. Consequently, EF Satsop fish experienced more consistent juvenile growth between sampling events than Newaukum River fish, whose growth appeared stagnant between August and December.

Within the Newaukum River, winter age-0/1 migrants (type II) were significantly smaller than spring age-1 smolts (type I) at the time of freshwater emigration, corroborating the result that type II fish migrated earlier and were behaviorally distinct from type I fish. Conversely, in the EF Satsop River, there was no difference in size at freshwater emigration between types I and II, although more than half of type II fish emigrated near the completion of annulus formation (in the last 25% of annulus material, S5 Fig.). This observation, combined with the later temporal completion of annulus formation we observed in juvenile samples, suggests most type II EF Satsop coho emigrated from freshwater close to, or during, spring. Despite no difference in size at emigration, EF Satsop type I fish were significantly smaller than type II fish at both the beginning and end of their first annulus, suggesting that individuals who enter and exit winter at a smaller size extend their stream rearing to continue growing and emigrate later. Studies documenting both fall/winter parr migrants, and spring age-1 smolts have noted little outmigration occurring in the interim time between pulses of these two emigrant groups [10,16,52]. This would suggest that EF Satsop late type II fish were part of the beginning of the spring smolt outmigration pulse, and that type I fish were slightly behind them in reaching some optimal/threshold outmigration size.

Our juvenile annulus timing validation was based on samples bridging a single winter season and it is possible that timing of annulus formation could vary between years. We expect water temperatures, and consequently annulus formation, in the rain driven Newaukum River to experience more interannual variation than in the groundwater driven EF Satsop. However, there are other factors that can also impact annulus formation timing (e.g., occupying side channel vs. mainstem habitat) and because of the water source difference, we would expect variation between streams to be greater than interannual variation within a stream.

Given the differences in timing of annulus formation and juvenile migration observed between the EF Satsop and Newaukum rivers, exploring these characteristics across additional years and other Grays Harbor subbasins (e.g., Johns, Wynoochee, Black, Skookumchuck, and Upper Chehalis Rivers) would be valuable for expanding our understanding of diversity and annual variation in life history strategies. Brennan et al. [4] demonstrated for Chinook (*O. tshawytscha*) and sockeye (*O. nerka*) salmon returning to the Nushagak River in Alaska, that the productivity of fish rearing among different habitats (i.e., different life histories) varies widely among years and across a range of spatial scales, ultimately stabilizing production of the entire basin. To our knowledge, all coho studies capable of determining the juvenile life history composition of returning adults have identified multiple successful strategies. Expanding the spatial scope of our work could provide insight into how life history diversity may influence productivity, stability, and resilience watershed wide [2–4,53].

Here we demonstrate that several successful juvenile life history strategies exist in Chehalis Basin coho salmon beyond the known spring emigration of age-1 fish. We add to the growing body of evidence that fall/winter migrants are not merely surplus production but can and do contribute to adult returns. We highlight how incorporation of previously unrecognized life histories that make meaningful contributions to a reproducing population could aid in addressing challenges with run size forecasting. Moreover, we emphasize that safeguarding the diversity of life history strategies will be crucial to the long-term productivity and stability of wild coho salmon in the Chehalis River watershed.

## Supporting information

**S1 Fig. Example scales from a wild (A) and a hatchery (B) age 1.1 EF Satsop coho salmon.** We used the size and pattern of the first annulus and freshwater zone to determine origin type. Scales from wild fish have a smaller and more defined first annulus than hatchery fish. Hatchery fish also consistently contain freshwater growth checks.
(TIF)

**S2 Fig. Individual otoliths from two juvenile coho salmon (*Oncorhynchus kisutch*) captured at age-1 in December of 2021 in the Newaukum River actively forming their second annulus.** Note these individuals would be age-2 as of January 1st 2022.
(TIF)

**S3 Fig. Otoliths from (A) juvenile and (B) adult coho salmon (*Oncorhynchus kisutch*) illustrating the more defined annuli observed in fish from the Newaukum River versus less defined annuli in fish from EF Satsop River individuals.**
(TIF)

**S4 Fig. (A) An adult otolith and (B) the corresponding otolith Sr:Ca for a coho salmon (*Oncorhynchus kisutch*) from the Newaukum River that was classified as life history type VI.** This fish emigrated from freshwater during formation of its first winter annulus and formed another two annuli in marine waters. Annuli spacing suggesting prolonged residence in brackish marine environment prior to its ocean migration.
(TIF)

**S5 Fig. Paired proportional histogram of life history type II fish showing the location of freshwater emigration within the first annulus.** The 0th percentile indicates the beginning and the 100th percentile indicates the end of the annulus.
(TIFF)

## Acknowledgments

We would like thank Dan Olson, Devin West, and crew at the Bingham Creek and Newaukum River juvenile traps who collected samples used in this study. We would like to thank Ellis "Sky" Cropper for help collecting juveniles by electrofishing in the Newaukum and EF Satsop Basins and Christina Jump for juvenile sample collection and data entry. We would also like to thank Lea Ronne, Curt Holt, Nick VanBuskirk, Justin Miller-Nelson, and staff of the Washington Department of Fish and Wildlife District 17 Management Team for adult coho collections in the Newaukum and Satsop rivers. Special thanks to the Bingham Creek Hatchery Staff and especially Joel Jaquez for aiding in sample collection at Bingham Creek Hatchery. Lastly, we would like to thank Adam Kent and Chris Russo from Oregon State University's Keck Collaboratory for Mass Spectrometry.

## Author contributions

**Conceptualization:** Austin J. Anderson, Andrew M. Claiborne, Marisa N. C. Litz.

**Data curation:** Austin J. Anderson, Andrew M. Claiborne.

**Formal analysis:** Austin J. Anderson, Andrew M. Claiborne.

**Funding acquisition:** Austin J. Anderson, Andrew M. Claiborne, Marisa N. C. Litz.

**Investigation:** Austin J. Anderson, Andrew M. Claiborne, Marisa N. C. Litz.

**Methodology:** Austin J. Anderson, Andrew M. Claiborne, Lance Campbell.

**Project administration:** Austin J. Anderson, Marisa N. C. Litz, Lance Campbell.

**Resources:** Austin J. Anderson.

**Supervision:** Andrew M. Claiborne, Lance Campbell.

**Validation:** Austin J. Anderson.

**Visualization:** Austin J. Anderson, Andrew M. Claiborne.

**Writing – original draft:** Austin J. Anderson, Andrew M. Claiborne.

**Writing – review & editing:** Austin J. Anderson, Andrew M. Claiborne, Marisa N. C. Litz, Lance Campbell.

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
