## [Decision Letter · Decision Letter 0]

12 Oct 2025

When did you leave home? Reconstructing juvenile life histories of Coastal Washington adult coho salmon (Oncorhynchus kisutch) using otolith structure and chemistry

PLOS ONE

Dear Dr. Anderson,

Thank you for submitting your manuscript to PLOS ONE. After careful consideration, we feel that it has merit but does not fully meet PLOS ONE’s publication criteria as it currently stands. Therefore, we invite you to submit a revised version of the manuscript that addresses the points raised during the review process.

https://journals.plos.org/plosone/s/submission-guidelines#loc-laboratory-protocols . Additionally, PLOS ONE offers an option for publishing peer-reviewed Lab Protocol articles, which describe protocols hosted on protocols.io. Read more information on sharing protocols at https://plos.org/protocols?utm_medium=editorial-email&utm_source=authorletters&utm_campaign=protocols .

We look forward to receiving your revised manuscript.

Kind regards,

Andrea Belgrano, Ph.D.

Academic Editor

PLOS ONE

Journal Requirements:

 1. You may seek permission from the original copyright holder of Figure(s) [#] to publish the content specifically under the CC BY 4.0 license.  

Additional Editor Comments:

I tend to agree with the comments and suggestions made by both reviewers.The manuscript is a timing and interesting study that will be very useful for providing novel information to be considered in guiding management a policy action towards conservation measured for coho salmon stocks/populations.

Reviewers' comments:

Reviewer's Responses to Questions

**Comments to the Author**

1. Is the manuscript technically sound, and do the data support the conclusions?

Reviewer #1: Yes

Reviewer #2: Yes

2. Has the statistical analysis been performed appropriately and rigorously?

Reviewer #1: Yes

Reviewer #2: Yes

3. Have the authors made all data underlying the findings in their manuscript fully available?

Reviewer #1: Yes

Reviewer #2: Yes

4. Is the manuscript presented in an intelligible fashion and written in standard English?

Reviewer #1: Yes

Reviewer #2: Yes

Reviewer #1: The research seeks to identify life history types of coho salmon in two tributaries to the Chehalis river in Washington state using the chronological properties of otoliths alongside chemical signatures indicative of marine entry or estuary use. Through these methods, the authors identify several life history types not accounted for in current fisheries management, suggesting additional life history types which may be contributing to a portfolio effect and uncertainty in run size estimates. The work is robust and considers several potential scenarios which may cause uncertainties in their own data. There is some room to add mention of additional uncertainties, however, the results presented here appear to be well supported and suggest a framework for similar monitoring of life history diversity using otolith chemical analysis. Specific suggestions and comments are uploaded as an attachment.

Reviewer #2: General comments

This manuscript reports important research that offers substantial insights for management and decision-making. While mostly well written there are several instances where the writing is overly vague, especially in the methods section. A light revision with a focus on improving the precision of language would strengthen the manuscript.

Because PLOS ONE is a general-purpose journal with a wide readership, the authors should not assume that all readers are familiar with otoliths. A short paragraph in the introduction explaining what otoliths are and how they grow would improve accessibility for a broader audience.

Research measuring Sr/Ca in otoliths often includes Ba/Ca to help differentiate estuarine from marine water. I am curious why barium was not measured here as it might have provided insight into the mystery surrounding the second and third annuli having a freshwater appearance. A quadrupole ICPMS should allow Sr, Ba and Ca to be analyzed simultaneously.

Line by line comments

Line 28: “…timing of otolith growth…” It is unclear what you are referring to here.

Line 31: The term “annulus measurements” is unclear. What did you measure on the annuli? Distance? Count?

Lines 93 and 96: Revise the sentence beginning at line 93 so that its first half stands as an independent sentence. Then combine the latter half of that sentence with the sentence beginning at line 96 to avoid repetition and improve the overall flow.

Line 138: Was this a carcass survey? Was it by boat, snorkel or foot?

Line 164: How are you delineating the quadrants?

Line 174: This paragraph can use clarification to better communicate your process. When discussing the aging of juvenile otoliths readers might think you are performing daily increment reads so specify that these are annual reads and you are differentiating age-0 from spring age-1. Portions of the methods that were used on both juveniles and adult otoliths should be independent of the paragraph dedicated to juvenile methods.

Line 177: Please specify which otolith structures you are referring to. How is assigned age different than [date of capture]-[Jan 1st]?

Line 183: How did the readers resolve disagreements? How many samples were resolved in this way?

Line 191: Change “corresponding Sr:Ca” to “corresponding Sr:Ca profile” or “corresponding Sr:Ca life history profile.”

Line 196: Expand on this sentence. You don’t need to define each individual life history type here but more detail than “A unique combination” would be helpful.

Line 214: Did you investigate other models for fork length back-calculation other a linear regression?

Line 237: Just R? Were any packages used?

Line 255-256: Rewrite the two sentences in these lines into one.

Line 271: Observing the relative contrast of the annuli between the two sites comes off as rather subjective. Image Pro has the functionality to measure luminosity to objectively compare contrast between the two study areas.

Line 300: Did you explain somewhere the difference between freshwater and marine annuli and is there a citation?

Line 309: Change “Sr:Ca” to “Sr:Ca profiles” or “Sr:Ca life history profiles”

Line 425: Add the word “is” between adjustment and needed.

**Do you want your identity to be public for this peer review?**  For information about this choice, including consent withdrawal, please see our Privacy Policy

Reviewer #1: **Yes: ** Ben Makhlouf

Reviewer #2: No

---

## [Author Response · Author response to Decision Letter 1]

21 Nov 2025

We have revised the manuscript as recommended by the reviewers and editor. The appropriate new manuscript files and response to reviewers have been uploaded in the attached files section. Thank you.

---

## [Editor Report · Decision Letter 1]

21 Dec 2025

When did you leave home? Reconstructing juvenile life histories of Coastal Washington adult coho salmon (*Oncorhynchus kisutch* ) using otolith structure and chemistry

PONE-D-25-42873R1

Dear Dr. Anderson,

We’re pleased to inform you that your manuscript has been judged scientifically suitable for publication and will be formally accepted for publication once it meets all outstanding technical requirements.

Kind regards,

Andrea Belgrano, Ph.D.

Academic Editor

PLOS One

Additional Editor Comments (optional):

The author(s) have fully addressed in the revised manuscript all the comments and suggestions that were made during the review process.

---

## [Editor Report · Acceptance letter]

PONE-D-25-42873R1

PLOS One

Dear Dr. Anderson,

I'm pleased to inform you that your manuscript has been deemed suitable for publication in PLOS One. Congratulations! Your manuscript is now being handed over to our production team.

Kind regards,

on behalf of

Dr. Andrea Belgrano

Academic Editor

PLOS One